# Clinical Application of Small Extracellular Vesicles in Gynecologic Malignancy Treatments

**DOI:** 10.3390/cancers15071984

**Published:** 2023-03-26

**Authors:** Fei Zheng, Jiao Wang, Dandan Wang, Qing Yang

**Affiliations:** Department of Obstetrics and Gynecology, Shengjing Hospital of China Medical University, Shenyang 110004, China

**Keywords:** small extracellular vesicles, gynecologic malignancies, isolation, detection, biomarker, therapy

## Abstract

**Simple Summary:**

Common gynecologic malignancies include ovarian, cervical, and endometrial cancer. Endometriosis is characterized by distant metastasis and is associated with gynecologic tumors. Most gynecologic malignancies are diagnosed in the middle and late stages of cancer, which complicates treatment; therefore, early diagnosis and therapy are beneficial to patient prognosis. Small extracellular vesicles (sEVs) have rich content, stable existence, and low antigenicity, which can be used for early diagnosis and precision treatment. However, current sEVs detection methods are difficult to popularize, and the contents of sEVs are complex and thus unsuitable to meet the requirements of clinical detection. This highlights the need to find more accurate and targeted molecules with diagnostic value. Our review aims to summarize the extraction and identification of sEVs and their application in the early detection and targeted therapy of gynecologic malignancies in recent years.

**Abstract:**

Small extracellular vesicles (sEVs) are the key mediators of intercellular communication. They have the potential for clinical use as diagnostic or therapeutic biomarkers and have been explored as vectors for drug delivery. Identification of reliable and noninvasive biomarkers, such as sEVs, is important for early diagnosis and precise treatment of gynecologic diseases to improve patient prognosis. Previous reviews have summarized routine sEVs isolation and identification methods; however, novel and unconventional methods have not been comprehensively described. This review summarizes a convenient method of isolating sEVs from body fluids and liquid biopsy-related sEV markers for early, minimally invasive diagnosis of gynecologic diseases. In addition, the characteristics of sEVs as drug carriers and in precision treatment and drug resistance are introduced, providing a strong foundation for identifying novel and potential therapeutic targets for sEV therapy. We propose potential directions for further research on the applications of sEVs in the diagnosis and treatment of gynecologic diseases.

## 1. Introduction

Ovarian cancer (OC), cervical cancer (CC), and endometrial cancer (EC) are the most common malignancies of the female reproductive system that seriously impact patient health. Endometriosis (EMS) shares its key characteristics with cancer and is a risk factor for OC and EC [1,2,3,4]. Therefore, identifying diagnostic and prognostic biomarkers and exploring novel therapeutic targets are imperative.

Cells communicate via cell-to-cell contact, secretory molecules, and extracellular vesicles (EVs). The International Society for Extracellular Vesicles (ISEV) defines EVs as particles naturally released by cells that are separated by lipid bilayers and cannot replicate. According to their size, they can be divided into small EVs (sEVs) (<100 nm or <200 nm), medium EVs, and large EVs (>200 nm) [5,6]. sEVs are heterogeneous vesicles that play important roles in intercellular communication. The lipid bilayer of sEVs allows encapsulation of various molecular cargoes, including DNA, mRNAs, miRNAs, and proteins, protecting them from degradation [7]. The release rates and contents of sEVs in healthy individuals differ from those in patients with cancer [8,9]. The nucleic acid content of sEVs largely consists of microRNAs (miRNAs), which represent the state of the parent cell [10]. Therefore, sEVs can act as novel biomarkers for diagnosis and prognosis. Long non-coding RNAs (LncRNAs), circular RNAs (circRNAs), and small RNAs derived from sEVs have attracted attention in the past decade as potential biomarkers and therapeutic targets for tumor treatment [11,12]. Liquid biopsy surveillance is performed to detect cancer biomarkers in body fluids, such as blood and urine [13]. As sEVs exist stably in body fluids, they have the potential for use as biomarkers for clinical applications in gynecologic malignancy diagnosis, prognosis, or therapeutic monitoring (Figure 1). The pathological states of sEVs can result in an increase in resistance to chemotherapeutic agents. The main mechanism by which sEVs participate in chemoresistance is the transfer of cargo from drug-resistant cells to drug-sensitive cancer cells, which enables the latter to develop resistance to the treatment [14]. Therefore, it is important to pay attention to how sEVs regulate the reaction of cancer cells to chemotherapy drugs. In addition, sEVs have been explored as drug carriers for targeted therapy (Figure 2). Isolation of sEVs in sufficient quantity and purity, based on their size, biochemical properties, and surface markers, is a prerequisite for their use in clinical settings [15].

This review describes the function of sEVs in the diagnosis and targeted therapy of gynecologic malignancies and summarizes the latest sEV isolation techniques. Through our conclusions, we provide theoretical support for the clinical development of sEV-based liquid biopsy and treatment strategies. Furthermore, we discuss the difficulties in using sEVs as biomarkers and for treating gynecologic tumors. Directions for further research on the applications of sEVs in the diagnosis and treatment of gynecologic diseases are proposed.

## 2. sEV Isolation and Detection

sEVs are typically less than 200 nm in diameter [6] and are produced in the extracellular space by nearly all cell types [26]. The number and content of sEVs are related to the state and type of the producing cell, which is caused by the distinct regulatory sorting mechanisms of the source cells [27]. The surface of sEVs is composed of a lipid bilayer that surrounds various surface proteins, including tetraspanins (CD81, CD63, and CD9), endosomal sorting complexes required for transport-associated components (TSG101 and Alix), integrins, lactadherin heat shock proteins (HSP60, HSP70, and HSP90), and antigen presentation proteins [28,29,30]. These proteins support the classification, selective recruitment, capture, and analysis of sEVs, in addition to certain disease-related proteins disseminated on the surface of sEVs that can be employed as biomarker in clinical diagnosis and prognosis [31]. However, sEVs have similar size and density to other body fluid components, such as lipoproteins [32]. Therefore, these components must be removed during the isolation of sEVs. However, there are still no standard terms for sEV subtypes based on biochemical components or physical characteristics [5].

Rapid and reliable isolation and detection of sEVs are important for their application in clinical diagnosis and treatment monitoring. The presence of various cellular and molecular components in bodily fluids hinder sEVs isolation, characterization, and clinical application [33]. The isolation of sEVs from biological samples can be difficult due to their small size, low concentration, and heterogeneity [30]. A standardized method for sEV separation and detection is yet to be established [34]. The common methods, basic principles, advantages, and disadvantages of sEVs isolation are summarized in Table 1. According to research findings, exosomes and plasma membrane-derived EVs may both be present in sEV-containing fractions and have a number of similar molecular components. Other subpopulations of sEVs exhibit functional behaviors that are equally relevant to be investigated as exosomes. The global secretion of different sEV components should not be viewed in line with the functional impacts of exosomes, but rather with that of sEVs in general. The term “exosomes” refers to the sEVs released during the exocytosis of multivesicular bodies containing intraluminal vesicles. ISEV endorses sEVs as the proper name for exosomes [5]. Therefore, the term “small extracellular vesicles (sEVs)” will be utilized in this review in accordance with the ISEV.

Microfluidics has been recently explored for the isolation of sEVs because it offers the advantages of using a lower quantity of chemicals, a high sensitivity, and a quick confirmation [33]. sEV detection approaches based on microfluidics include fluorescence imaging [65], colorimetry [66], surface plasmon resonance [67], optical special effect methods, magnetic and electrochemical detection [33], quartz crystal microbalance (QCM) [68], and Raman spectroscopy [69]. A magneto-fluorescent sEV nanosensor was established to separate and analyze tumor-derived sEVs in blood based on their unique membrane proteins [70]. A novel 3D nanopatterned microfluidic chip that can detect circulating sEVs with extremely high sensitivity would be appropriate for profiling sEVs in different cancers [35]. However, there has been limited progress in improving the sensitivity, purity, speed, and multiplicity of microfluidics-based techniques [33,71,72]. sEVs in the plasma of patients with malignancy contain substantial amounts of tumor-derived phosphatidylserine in their outer membrane leaflet, which can aid cancer diagnosis [73]. Sharma et al. [74] developed a highly sensitive enzyme-linked immunosorbent assay-based system to detect picogram amounts of sEV phospholipids as cancer biomarkers in the plasma. A portable electrochemical aptamer sensor combines the advantages of electrochemical methods, DNA nanostructures, and DNA aptamers with expanded nucleotides for the rapid and direct detection of hepatocyte-derived sEVs, laying the foundation for quantifying sEVs in complex body fluids [75]. The efficiency of the single-bead encapsulation method was increased to 86% owing to the bead ordered arrangement droplet (BOAD), which was the first approach to accomplish the fastest ordered arrangement of particles in the shortest structure. Thus, BOAD can be used for single sEV detection [76]. Raman spectroscopy is a dependable, quick, and operator-independent method that only requires a small number of EV samples. However, the method lacks data on the distribution of particle sizes, certain biomarkers, and the EV structures [69]. Currently, there are few methods available for quantifying EVs. The QCM-based biosensing approach can quantify desiccated EVs, even with a small amount of liquid in a brief amount of time [77]. QCM combined with the Adaptive Interaction Distribution Algorithm can be applied to illustrate the kinetics between affinity ligands and interactions between EV subpopulations [78]. QCM with electrochemical impedance spectroscopy effectively captures sEVs via the surface protein CD63 [79].

Microfluidic techniques for isolating sEVs are founded on physical characteristics or immunoaffinity [80]. Physical property-based microfluidic techniques for sEV isolation rely on filtration, deterministic lateral displacement, acoustic wave, electrical field, and viscoelastic flow characteristics [80]. Immunoaffinity-based microfluidics is anchored to the principle of antigen–antibody interactions. Microfluidic devices can efficiently capture sEVs as biomarkers for the early detection of high-grade serous OC [81]. However, heterogeneous populations of sEVs exist despite the EV classification criteria [82,83]. A previous study identified two sEV subpopulations via asymmetric-flow field-flow fractionation (AF4) [84], proving that AF4 can be used as an enhanced analytical tool for isolating and addressing the complexities of heterogeneous nanoparticle subpopulations. The online coupled immunoaffinity chromatography-AF4 (IAC-AF4) can be employed in the clinical setting to produce EVs subpopulations and analyze the amino acids and glucose of EV subpopulations from human plasma [46]. Furthermore, the IAC-AF4 is appropriate for other ligand that can be paired to the monolithic disk column for research. ExoTIC is a size-based separation tool that can be used on various samples, including cell culture medium, plasma, and urine. ExoTIC can wash away free proteins and nucleic acids and rapidly isolate large amounts of sEVs from small quantities of blood [85].

The establishment of a multifunctional, reliable, and efficient sEV isolation and detection method is the key to developing sEV-based, non-invasive diagnostic and therapeutic approaches. Furthermore, combining multiple methods and principles can help standardize an optimal technique to isolate sEVs. For example, microfluidic chips can be utilized to separate subpopulations of EVs through an integrated approach combined with electrophoresis and dielectrophoresis [45,54,86] based on different electrophoretic mobilities. Due to its selectivity and accuracy, this combination has proven to be a viable application tool in the field of particle separation [87]. Meanwhile, other charge-based techniques have also been applicable to isolate and concentrate EVs [88]. Simple separation methods for small-volume samples have been developed to overcome the shortcomings of ultracentrifugation (UC), which is more suitable for large-volume clinical samples [89]. When UC is combined with ultrafiltration (UF), the UF membrane is used to screen large EVs and cells; UC subsequently separates sEVs from proteins [90]. Microfluidic technology can achieve highly efficient and sensitive enrichment, separation, and multi-information detection of sEVs in a single chip.

Physical property-based microfluidics can isolate sEVs at a low cost. Yield, purity, and cost-effectiveness are important parameters for evaluating the efficiency of sEV extraction methods. In summary, each method has its own advantages and disadvantages; thus, extraction methods should be selected for targeted research according to their specific characteristics.

## 3. sEV and OC

OC is a malignant gynecologic tumor with a high mortality rate. Given late clinical presentation, OC is diagnosed only after its progression to advanced stages. In addition, it is often chemo-resistant and has a high recurrence rate. However, early-stage OC is curable in 90% of women [91], highlighting the need to develop novel methods for early diagnosis and treatment of OC.

### 3.1. Biomarkers

Liquid biopsy is minimally invasive and allows convenient serial measurements during treatment. However, individual approaches with high specificity and sensitivity for ovarian tumor detection are currently lacking [92]. The sEVs derived from different donor cells differ in their characteristics and cargo [93]. Therefore, the sEV miRNAs of patients with benign ovarian diseases differ from those with malignant ovarian diseases [94].

OC prefers peritoneal cavity invasion through ascites; therefore, sEVs are detectable in the ascites and blood of OC patients, highlighting their potential for use in less invasive diagnostic approaches and as potential targets for early targeted therapy [95,96,97]. Circulating cell-free miRNAs and sEVs are the main components of liquid biopsies [98], and ascites contain tumor-associated sEVs. The expression of miR-4732-5p [99], miR-205 [100], and miR-200b [101] are upregulated in plasma sEVs of patients with OC; thus indicating these miRNAs as candidate diagnostic biomarkers for OC. Metastasis-associated lung adenocarcinoma transcript 1 (MALAT1) is a lncRNA connected with cancer angiogenesis. The high expression of MALAT1 derived from plasma sEVs is highly related to metastasis of OC and can be used as a biomarker for the prognosis of OC [18]. Genomic DNA (gDNA) is mostly restricted to the nucleus; however, recent studies have found gDNA and other nuclear contents in sEVs [102]. In addition, gDNA is shuttled into multivesicular bodies via tetraspanins. sEV gDNAs in OC patients reflect the copy number variation status of the primary tumor, exposing instructive DNA alterations [102]. sEVs containing protein signatures specific to OC have been isolated from the ascites [103,104] and serum [35] samples of OC patients. Nanog is secreted and detected only in high-grade serous carcinoma sEVs in effusions and modulates tumor-promoting cellular processes in vitro [105]. A 3D-nanopatterned microfluidic chip was used for the quantitative detection of circulating sEV CD24, epithelial cell adhesion molecules [106,107], and folate receptor alpha (FRα) [35] as OC biomarkers. The results showed that sEV FRα considerably distinguished early-phase OC from that of the terminal stage, whereas CD24 and EpCAM could not [35].

### 3.2. Treatment

In addition to their potential as biomarkers, sEVs could be used as therapeutic targets. Cisplatin is a first-generation platinum-based drug that can directly interact with cancer cell DNA to prevent DNA replication and transcription [108]. The miRNA of sEVs alters OC resistance by controlling targeted gene expression through various mechanisms. Paclitaxel resistance is brought on by sEVs miR-1246, which controls the Cav-1/p-gp/M2 axis [109]. A dual strategy for efficient chemosensitization and anticancer therapy in OC patients involves the use of miR-1246 inhibitors in conjunction with paclitaxel [109].

sEVs are natural nanosized vesicles and have immense potential for drug delivery [24]. miR-497/TP-HENPs are bioinspired hybrid nanoparticles characterized by nano size, good drug encapsulation effectiveness, and protection of nucleic acids, which can stably and continuously release miR-497 and suppress OC chemotherapy resistance both in vivo and in vitro [110]. Tumors cannot develop more than 1–2 mm without a blood supply [111]. Anti-angiogenic therapeutic strategies may be beneficial in OC treatment [112]. Notably, miR-484 simultaneously inhibits the level of vascular endothelial growth factor (VEGF)-A in cancer cells and its associated receptors in endothelial cells. The RGD-modified sEV miR-484 has been shown to enhance vascular normalization, increase the susceptibility of cancer cells to chemotherapy, and increase the life of tumor-bearing mice after chemotherapy [113]. Soluble E-cadherin (sE-cad) derived from E-cadherin [114] was overexpressed in the ascites and serum of OC patients and predicted poor prognoses [115]. sE-cad-positive sEVs mediate heterophilic connections with vascular endothelial-cadherin and transmit a new sequential activation of NF-κB signaling and β-catenin. Similar to VEGF, sE-cad can efficiently induce angiogenesis but without VEGF [116]. Furthermore, targeting sE-cad may be advantageous compared to targeting VEGF, as sE-cad is less susceptible to the resistance mechanisms of cancer cells [117].

Immune cell evasion and infiltration could be regulated by stimulating or inhibiting the release of sEVs [118]. Epithelial OC triggers macrophage recruitment and induces them to develop a tumor-associated macrophage (TAM)-like phenotype [119,120]. Notably, miR-223-enriched sEVs from hypoxic macrophages can improve the [121] malignant phenotype and chemotherapy resistance of epithelial OC through the PTEN-PI3K/AKT pathway [122]. Inhibiting miR-223 expression is insufficient to eliminate chemotherapy resistance induced by the sEVs produced by TAMs [122]. Therefore, other factors contributing to sEV-induced chemoresistance need to be elucidated. The sEVs-mediated T cell inhibition occurs via phosphatidylserine + sEVs binding to the phosphatidylserine receptor, although the suppression is reversible and transient. Once the sEVs have been removed, sEV-exposed T cells can become active again [123]. However, other studies have shown that this effect is irreversible 1–4 days following exposure [124,125]. The immunotherapeutic properties of sEVs should be considered as a new research direction for tumor immunotherapy [23].

## 4. sEV and CC

Although highly preventable, CC is a frequent cause of cancer and cancer-related mortality in women. Most CC cases are caused by infection with a subtype of the human papilloma virus (HPV). The treatment modalities for primary CC include surgery, radiotherapy, and concurrent chemoradiotherapy, with recurrence rates ranging from 25 to 61% after initial treatment [126]. Tumor-derived sEVs have been detected in patients with CC [127]. The functions of CC-derived sEVs in clinical diagnosis and therapy are discussed below.

### 4.1. Biomarkers

Plasma miRNAs are potential biomarkers of CC. The expression of miR-21-5p and miR-146a-5p are upregulated in CC tissues, whereas those of sEV miR-151a-3p, miR-146a-5p, and miR-2110 are upregulated in the plasma [128]. miR-1468-5p is overexpressed in serum sEVs of CC patients and is known to be connected with PD-1^+^ CD8^+^ T cells and PD-L1^+^ lymphatics; thus, it is clinically relevant to CC prognosis [129]. The expression level of sEV miR-125a-5p in CC patients is significantly lower than that in healthy control patients [130]. Overexpression of serum-derived sEVs has been shown to be closely associated with aggressive clinical features and poor prognosis in CC patients; incidentally, lncRNA DLX6-AS1 levels in the blood decreased significantly 90 days after treatment [131]. Therefore, signature non-coding RNAs (ncRNAs) extracted from peripheral blood are a potential diagnostic biomarker for CC.

### 4.2. Treatment

sEVs play a crucial role in the inflammatory microenvironment of cancers [132]. sEVs originating from HPV-infected cells can mediate the diversion of mRNAs, miRNAs, and cytokines between cells [133], whereas those secreted by HPV-positive cells can enhance virus-induced tumorigenesis by transferring E6 and E7 oncogenes and HPV-deregulated miRNAs to additional target cells in the extracellular environment [134]. Dysregulating miRNA expression is not specific to HPV genotypes [135] and has the ability to regulate the initiation and progression of inflammation-induced CC [136]. miRNA-based therapies rely on the administration of miRNA mimetics. To date, only a small number of tumor-inhibitory miRNAs, such as miR-34a, miR-143/145, and miR-125, have been suggested as potential therapeutics. The off-target or non-specific effects that could reduce treatment effectiveness may pose additional constraints [133]. sEV treatment downregulates the expression of miR-1231 whilst upregulating the expression of IFNAR1 and LINC000673, thereby suppressing CC [137,138]. Zhu et al. injected sEVs isolated from patients with the rs11655237 SNP genotyped as GG, GA, and AA in a mouse model of CC and found that sEVs inhibited tumor growth in vivo [138]. The ability of sEVs to induce drug resistance is determined by their cargo during intercellular communication. The miR-22 expression level is significantly increased in CC cells. When miR-22 encapsulated in sEVs is administered to CC cell lines, the levels of c-Myc binding protein and human telomerase reverse transcriptase significantly decrease with increased radiosensitivity [139]. CC-secreted sEVs carrying lncRNA HNF1A-AS1 (a competing endogenous RNA of miR-34b) promote Tuftelin 1 expression, which increases the cisplatin resistance of CC cells [140]. Cancer-associated fibroblast (CAF)-secreted sEV miR-1323 leads to abnormal upregulation of miR-1323 in CC cells. The up-regulated miR-1323 promotes the proliferation, invasion, migration, and radio-resistance of CC by targeting PABPN1 and activating the Wnt/β-catenin signaling pathway [141]. miR-651 was down-regulated in cisplatin-resistant CC cells. miR-651 from the sEVs of cancer cells restricts cisplatin resistance and development and directly targets ATG3 in CC [142].

A crucial step in the development of cancer is local angiogenesis [143]. As a necessary component of tumor growth, angiogenesis is a prospective therapeutic target for advanced CC [144]. miR-221-3p is transferred from cancer cells to vascular endothelial cells by CC sEVs, which also suppresses the expression of THBS2 and encourages angiogenesis [145]. Research has shown that sEV miR-221-3p from cutaneous squamous cell carcinoma can compensate for the disadvantages of anti-angiogenesis therapies mostly targeting the VEGF axis [146], such as varied responses, potential drug resistance, and non-negligible toxicity [147]. miR-221-3p is intimately linked to peritumoral lymphangiogenesis and lymph node metastases [148]. CC cells promote innervation by releasing sEVs that stimulate neurite outgrowth. Compared to HPV-negative cell lines, HPV-positive cell lines are more successful at promoting neurite outgrowth. As microenvironmental factors, nerves may contribute to tumor progression and poor clinical outcomes [149,150,151]. sEV-mediated innervation or tumor-associated neurons are possible therapeutic targets to enhance outcomes in this patient population [152]. Thus, taken together, sEVs are a novel drug delivery system for CC radiotherapy.

## 5. sEV and EC

EC is the most frequently occurring gynecologic cancer in high-income nations, and its global prevalence is increasing [153]. Predominantly occurring in postmenopausal women, obesity is the principal cause of increasing EC prevalence. If detected at an early stage, the 5-year survival rate after surgical treatment is >90% [154]. Therefore, finding a highly specific and sensitive EC diagnostic tool is important for improving EC patient prognoses.

### 5.1. Biomarkers

The identification of novel sEVs-related liquid biopsy molecules aids in the early detection of EC. For example, the expression levels of miR-15a-5p [155], miR-20b-5p [156], and miR-151a-5p [157] are expressively upregulated in the plasma-derived sEVs of EC patients. miR-15a-5p expression is considerably more abundant and tissue-specific in EC tissues than that in cervical, breast, ovarian, and lung cancer tissues. Therefore, plasma-derived sEVs are potential diagnostic biomarkers for the early detection of EC. Women with polycystic ovary syndrome (PCOS) have a 2.7-fold greater chance of developing EC. miR-27a-5p is the upregulated miRNA in the plasma sEVs of patients with PCOS and stimulates EC cell invasion and migration [158]. EV-based biomarkers have the potential for use in EC liquid biopsies. Moreover, the expression level of circRNAs is higher in EVs derived from the sera of patients with EC than in those from the sera of the healthy controls [159]. Abundantly expressed miR-200c in urinary sEVs is a potential preliminary diagnostic marker for EC [160]. Potential EC indicators, such as decreased miRNA-10b and miRNA34-b, can be found in EVs extracted from the peritoneal lavage of EC. [161]. Decreased miRNA-10b expression in EC tissue has shown a significant association with decreased overall survival [162]. EVs containing proteins, lipids, and DNA involved in intercellular communication are alternative forms of liquid biopsy [163]. A cohort of EC patients with a high risk of recurrence was shown to have the adhesion protein LGALS3BP substantially enriched in their circulating EVs [164]. Furthermore, high levels of ANXA2 in EVs extracted from the peripheral blood of EC patients have shown correlations with tumor histology, grade, stage, and recurrence risk [165].

The most typical liquid biopsies are plasma, serum, and urine biopsies [166]; however, uterine fluid in EC provides a better illustration of the molecular alterations occurring in the tumor [166,167]. Uterine fluid-derived sEVs and the non-RNA content of sEVs are rarely studied and therefore require further exploration.

### 5.2. Treatment

sEVs released by EC may be transported to normal endometrial cells. sEV miR-499 functions as a tumor inhibitor in EC development and angiogenesis though regulating VAV3 [168]. miR-26a-5p-devoid sEVs absorbed by human lymphatic endothelial cells could induce lymphatic vessel formation [169]. sEV miR-133a regulates FOXL2 down-regulation in EC and can be delivered to normal endometrial cells. Survival analysis indicates that FOXL2 might play an important role in the progression of EC [170]. In addition, sEVs from other sources affect EC development. For example, plasma sEVs containing LGALS3BP contribute to EC growth, progression, and angiogenesis [171]. sEVs miR-192-5p produced by TAMs is overexpressed and clearly inhibits tumor growth by regulating EC cell apoptosis and epithelial-mesenchymal transition [172]. In a previous study, TAM-derived sEVs transferred hsa_circ_0001610 to EC cells, and the overexpressed hsa_circ_0001610 released cyclin B1 expression through adsorbing miR-139-5p, thereby weakening the radiosensitivity of EC cells [173]. CD8^+^ T cell-derived sEV miR-765 obtained from the peripheral blood of healthy human donors [174] and miR-503-3p derived from the sEVs of human umbilical cord blood mesenchymal stem cells [175] can considerably inhibit EC cell growth. *Aurea helianthus* extract-inducing sEVs accelerate cellular senescence by inhibiting DNA repair in EC cells [176]. These findings could suggest a valid molecular target for EC therapy with the use of sEVs as a new treatment target.

## 6. sEV and EMS

Endometrial tissue that is present outside of the uterus is a defining feature of a persistent gynecologic disorder known as EMS. Between 6–10% of women of reproductive age are affected by this disease [177], with primary symptoms including pelvic pain and infertility. Despite its high prevalence, EMS is often not diagnosed until after several years and is prone to misdiagnosis, thereby affecting the treatment [178]. Prompt identification and therapy are therefore essential for efficient EMS clinical management. EMS is a benign disease; however, it is characterized by distant metastasis and leads to an increased risk of gynecologic malignancy [179,180]. Notably, sEVs play a role in metastasis in various diseases. In the next section, we discuss their application in the detection and treatment of EMS.

### 6.1. Biomarkers

sEVs are functionally relevant to the pathophysiology of EMS [181], as sEV levels in the peritoneal fluid (PF) change according to the illness stage and cycle phase [182]. Five proteins are unique in EMS: (fragments of) peroxiredoxin-1 (PRDX1), annexin A2 (ANXA2), histone H2A type 2-C, inter-α-trypsin inhibitor heavy chain H4 (ITIH4), and tubulin a-chain. The proteins particularly present in sEVs from patients with EMS could serve as disease biomarkers; however, their specificity and accuracy remain to be confirmed [183]. In addition, heat shock protein levels have been shown to be increased in sEVs released by endometriotic cells [184]. Moreover, plasma-derived EVs transfer miR-27a-3p, miR-375, and miR-30d-5p specific to EMS [185]. This knowledge could be useful in developing novel diagnostic tools for patients with gynecologic pathologies.

### 6.2. Treatment

Uterine endothelial cells promote miR-138 expression to induce sEV-mediated inflammation and apoptosis in EMS via the VEGF/nuclear factor (NF)-κB signaling pathway [186]. Treatment with an NF-κB or VEGF inhibitor promotes cell proliferation and reduces inflammation in a co-culture of uterine endothelial and THP-1 cells following miR-138 downregulation [186]. miRNA-30d is secreted by the human endometrium and packed into derived sEVs [187] and is a known plasma-circulating inflammation-related miRNA [188]. Based on previous studies, miRNA-30d deficiency in EMS patients with plasma EV may be related to dysregulated chronic inflammation [185] and promote angiogenesis by targeting the MYPT1/cJUN/VEGF-A pathway [189]. Upregulated miR-214 delivered by ectopic endometrial stromal cell-derived sEVs has been shown to reduce connective-tissue growth factor and collagen αI expression in stromal and endometrial epithelial cells and inhibit fibrogenesis [190]. The mRNA expression profile of sEVs extracted from endometriotic cell culture supernatants reflects the pathobiology of EMS. LncRNA and miRNA networks have been implicated in tumorigenesis, proliferation, invasion, and hypoxia-induced angiogenesis [191]. Collectively, these findings propose novel candidates that may serve as therapeutic targets for treating EMS.

In summary, sEVs play a crucial role in gynecologic malignancy diagnosis and treatment. For example, sEV miRNAs have been shown to be reliable biomarkers for early detection and can predict treatment response [6]. Clinical diagnosis of gynecologic malignancy using sEVs has been extensively investigated (Table 2). The function of sEVs in gynecologic malignancy treatment includes therapeutic targeting and drug delivery. Generally, they can serve as therapeutic targets of chemotherapeutic agent resistance (Table 3). Overall, sEVs have the potential to revolutionize cancer diagnosis and therapy. Further research is required to fully comprehend the mechanisms underlying sEV function in cancer and develop more effective sEV-based diagnostic and therapeutic strategies.

## 7. Conclusions

sEVs are highly heterogeneous in the context of their physical characteristics and cargo components. Cancer cell-derived sEVs carrying cancer cell-specific DNA, RNA, proteins, lipids, glycans, and other molecular substances are released into blood, urine, saliva, ascites, and even cervicovaginal fluid, suggesting their potential as cancer biomarkers [214]. The cell-specific proteins and genetic material in sEVs reflect the origin and physiological state of cells and can be explored as preclinical biomarkers for cancer. As important regulatory molecules, miRNAs participate in crucial life processes, including viral defense, hematopoiesis, organ formation, cell proliferation and apoptosis, fat metabolism, and tumorigenesis [215]. Although these advantages highlight the potential of tumor-derived sEVs as diagnostic biomarkers, their clinical application needs further investigation and validation owing to the lack of standardized and effective methods for isolation and identification.

sEVs provide important signal molecules involved in transduction mechanisms in gynecologic malignancies to regulate angiogenesis, neurogenesis, immune dysfunction, inflammation, migration, and invasion; thus, showing potential for use as therapeutic targets. Moreover, sEVs can overcome natural barriers owing to their inherent cell-like properties. Given their nanoscale size, excellent biocompatibility, long half-lives, potential ability to express targeted ligands, and natural ability to carry macromolecules, sEVs can be utilized as drug delivery vehicles for cancer therapy [216]. Furthermore, sEVs exert immune-activating and immunosuppressive effects in cancer. Effective treatment can be achieved by interfering with the secretion and uptake of sEVs or even eliminating sEVs, and the potential utility of sEVs as cancer therapeutic targets has become the focus of ongoing research.

Cargos isolated from circulating and PF sEVs may be valuable for the diagnosis of gynecologic malignant diseases and act as novel therapeutic targets. Most sEVs-related studies have evaluated their role in OC; however, similar studies are lacking for other diseases. Thus, future research should focus on the role of sEVs in various cancers. The main obstacle in the diagnosis of sEVs is that the isolation technology is difficult to popularize. Therefore, attention should be paid to the development of economic, rapid, and accurate detection technology to avoid the influence of non-tumor sEVs. In addition, the sensitivity and specificity of diagnostic markers still require verification in a large number of clinical patients. Additionally, newly available tools and advanced technologies should be employed to identify promising biomarkers and therapeutic targets for various diseases including cancer to reduce the incidence of chemoresistance and adverse drug reactions.

## Figures and Tables

**Figure 1 cancers-15-01984-f001:**
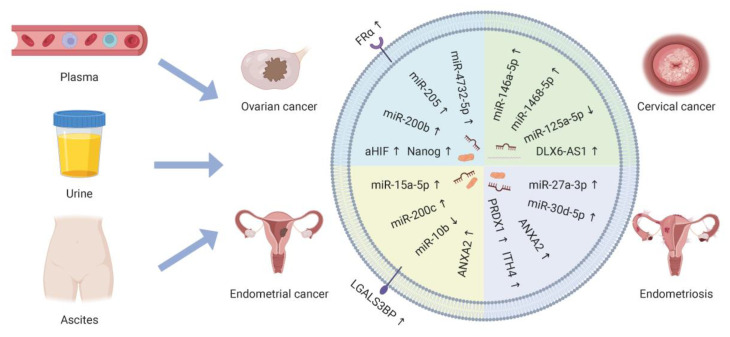
Sources and contents of small extracellular vesicles (sEVs) for liquid biopsy in gynecologic malignancies. These contents are potential biomarkers that can provide information about tumor progression, prognosis, and response to therapy [7]. The up or down arrows indicate the increase or decreasein sEV contents in the corresponding pathological states, respectively (Created with https://www.biorender.com/ (accessed on 26 February 2023)).

**Figure 2 cancers-15-01984-f002:**
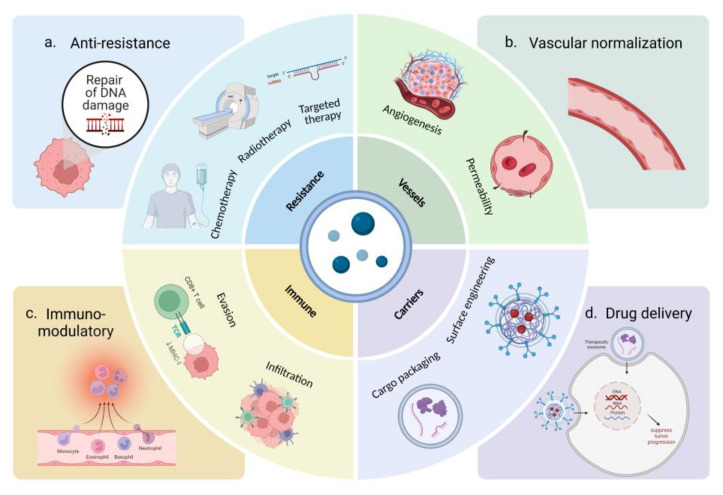
Principles and applications of sEVs in cancer therapy. (**a**) sEV contents induce chemo [14], radiation [16], and targeted therapy resistance [17] by changing the sensitivity of recipient cells through intercellular communication; (**b**) sEVs can regulate angiogenesis [17,18] and vascular permeability [19,20] by influencing the behavior of endothelial cells that line blood vessels; (**c**) sEVs can regulate the evasion [21] and infiltration [22] of immune cells by stimulating or suppressing immune responses, that is, they have the ability to act as immunomodulators [23]; (**d**) direct incubation of sEVs with cargo [24] or sEVs with surface engineering [25] can achieve the function of therapeutic cargo packing and targeted therapy (Created with https://www.biorender.com/ (accessed on 26 February 2023)).

**Table 1 cancers-15-01984-t001:** Techniques for sEV isolation.

Principle	Method	Advantages	Disadvantages	Ref.
Size and density	Differential ultracentrifugation	High purity, high sample volumes	High instrumental cost, time consuming, large samples	[35,36]
	Density gradient centrifugation	High purity	Time consuming, structural damage, large samples	[37,38]
	Ultrafiltration	Fast, simple operation, cheap equipment	Membrane clogged easily, lack specificity	[39,40]
	Size-exclusion chromatography	High purity, intact sEVs, low sample	Low efficiency, lack specificity	[41,42,43]
	Flow field-flow fractionation	High purity, high efficiency, provide information on sEVs, maintain protein activity, obtain subpopulations	Sample pretreatment	[44,45,46]
Solubility alteration	Polymer precipitation	High efficiency, intact sEVs	Low purity	[47,48]
Immune specificity	Enzyme-linked immunosorbent assay	High purity	Sample pretreatment	[49,50]
	Magneto- Immunoprecipitation	Keep protein activity, simple operation, distinguish source of sEVs	High cost	[51,52,53]
	Immunoaffinity chromatography with polymeric monolithic disks	Fast, simple operation, high throughput, obtain subpopulation, separate large biomolecules	Specific purpose-made instrumentation	[45,46,54]
Microfluidics	Physical properties	High efficiency, simple operation	Impurities	[55,56]
	Immune specificity	High efficiency, high throughput	Complicated sample pretreatment	[57,58]
Charges	Ion-exchange	Fast, high purity, intact sEVs	Sample pretreatment, simple sample	[59,60,61,62]
	Electrophoresis and Dielectrophoresis	Provide information on sEVs, obtain subpopulations, distinguish source of sEVs	Sample pretreatment, medium system integration and portability	[63,64]

**Table 2 cancers-15-01984-t002:** sEV contents serving as diagnostic biomarkers in gynecologic diseases.

Disease	Content	Expression	Source	Ref.
OC	miR-4732-5p	Up	Plasma	[99]
	miR-205	Up	Plasma	[100]
	miR-200b	Up	Plasma	[101]
	miR-34a	Up	Plasma	[192]
	miR-375 and miR-1307	Up	Plasma	[193]
	miR-200a-3p, miR-766-3p, miR-26a-5p, miR-142-3p, let-7d-5p, miR-328-3p, miR-130b-3p and miR-374a-5p	Up	Plasma	[194]
	miR-106a-5p, let-7d-5p, and miR-93-5p; miR-122-5p, miR-185-5p, and miR-99b-5p	Up; down	Plasma	[195]
	miR-21, miR-141, miR-200a, miR-200c, miR-200b, miR-203, miR-205, and miR-214	Up	Plasma	[94]
	circFoxp1	Up	Plasma	[196]
	circ-0001068	Up	Plasma	[197]
	gDNA	Up	Plasma and ascites	[102]
	SOX2 and SOX9	Up	Effusions	[105]
	FRα	Up	Plasma	[35]
CC	miR-146a-5p, miR-151a-3p, and miR-2110	Up	Plasma	[128]
	miR-125a-5p	Down	Plasma	[130]
	let-7d-3p and miR-30d-5p	Down	Plasma	[198]
	miR-21 and miR-146a	Up	Cervicovaginal lavage	[199]
	miR-1468-5p	Up	Plasma	[129]
	lncRNA-EXOC7	Up	Plasma	[200]
	lncRNA DLX6-AS1	Up	Plasma	[201]
EC	miR-15a-5p	Up	Plasma	[155]
	miR-20b-5p	Up	Plasma	[156]
	miR-151a-5p	Up	Plasma	[157]
	miR-200c-3p	Up	Urine	[160]
	miR-383-5p, miR-10b-5p, miR-34c-3p, miR-449b-5p, miR-34c-5p, miR-200b-3p, miR-2110, and miR-34b-3p	Down	Peritoneal lavage	[161]
	circ-0109046 and circ-0002577	Up	Plasma	[159]
	LGALS3BP	Up	Plasma	[171]
EMS	miR-30d-5p, miR-16-5p, and miR-27a-3p	Unique	Plasma	[185]
	PRDX1, H2A type 2-C, ANXA2, ITIH4, and the tubulin α-chain	Unique	Plasma	[182]

**Table 3 cancers-15-01984-t003:** sEV contents serve as therapeutic targets for resistance in gynecologic malignancies.

Disease	Content	Target/Pathway	Source	Function	Ref.
OC	miR-1246	Cav-1/P-gp/M2-type oncogenic macrophages	OC cells	Promote paclitaxel resistance	[109]
	miR-21	APAF1	Cancer-associated adipose cells or CAFs	Suppresses apoptosis; promote paclitaxel resistance	[202]
	miR-21-3p	NAV3	OC cells	Promote cisplatin resistant	[203]
	miR-21-5p	PDHA1	OC cells	Promote glycolysis; inhibit cisplatin resistance	[204]
	miR497/TP-HENPs	PI3K/AKT/mTOR	OC cells and hybrid nanoparticles	Restrain cisplatin resistance; induce M2 to M1 polarization of macrophages	[110]
	miR-891-5p	MYC/CNBP	OC cells	Promote carboplatin resistance	[205]
	miR-484	VEGF-A	RGD-modified sEVs	Improve vascular normalization; sensitize to chemotherapy	[113]
	miR-429; miR-3	NF-κB; CASR/STAT3	OC cells	Promote cisplatin resistance	[206]
	miR-223	PTEN-PI3K/AKT	Hypoxic macrophages	Promote multidrug resistance	[122]
	miR-146a	LAMC2	OC cells	Promote docetaxel and taxane resistance	[207]
	miR-4315	Bim	OC cells	Promote anti-PD1	[208]
CC	miR-22	MYCBP/hTERT	CC cells	Promote radiosensitivity	[139]
	miR-106a/b	SIRT1	Cisplatin resistant hepatocarcinoma cells	Promote cisplatin sensitivity	[209]
	miR-320a	MCL1	Engineered sEVs	Restrain cisplatin resistance	[210]
	miR-651	ATG3	CC cells	Restrain cisplatin resistance	[142]
	miR-1323	PABPN1/Wnt/β-catenin	CAFs	Promote radioresistance	[141]
	circ-0074269	miR-485-5p/TUFT1	CC cells	Promote cisplatin resistance	[211]
	lncRNA PDHB-AS	RBMX	CC cells	Restrain cisplatin resistance	[212]
	lncRNA MALAT1	miR-370-3p/STAT3/PI3K/Akt	CC cells	Promote cisplatin resistance	[213]
	lncRNA HNF1A-AS1	miR-34b/TUFT1	CC cells	Promote cisplatin resistance	[140]
EC	Circ-0001610	miR-139-5p/cyclin B1	M2-polarized macrophages	Restrain radiosensitivity	[173]

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
