# Peer review of "Clinical Application of Small Extracellular Vesicles in Gynecologic Malignancy Treatments"

_cancers, 2023, doi:10.3390/cancers15071984_

Round 1
Reviewer 1 Report
The authors summarized the current knowledge on the use of exosomes in the treatment of urogynecological disorders.
The manuscript requires the following corrections:
1. There is an error in the paper title (clinical, not clinical).
2. Figures 1 and 2 should precisely describe what they represent. In addition, they are instead a summary of the review, not its introduction.
3. The authors should first characterize what EVs are in general and then move on to describing exosome isolation methods.
4. The proper name for exosomes is sEVs.
5. The authors described exosomes as biomarkers of several female diseases and how they modulate the treatment of these diseases. Mostly, it is an increase in resistance to chemotherapeutic agents. This aspect is weakly presented in the paper made by the authors.
6. The text lacks tables that could help understand complex molecular relationships.
7. References must be extended with the following papers:
doi: 10.3390/cells11182913.
doi: 10.3390/cells8070727.
doi: 10.3390/cells8020099.
doi: 10.3390/cells12030356.
doi: 10.3390/cancers15010082.
doi: 10.3390/cancers12123563.
doi: 10.3390/cancers12102825.
doi: 10.3390/life11101044.
Reviewer 2 Report
This review focuses on the role of exosomes in the diagnosis and targeted therapy of gynecologic malignancies, including ovarian cancer, cervical cancer, and endometrial cancer. Exosomes have also been explored as drug carriers for targeted therapy. However, isolating exosomes in sufficient quantity and purity is a prerequisite for their clinical use. The review summarizes the latest exosome isolation techniques and discusses the difficulties in using exosomes as biomarkers and for treating gynecologic tumors. The authors propose directions for further research on the applications of exosomes in the diagnosis and treatment of gynecologic diseases.
Line 77: Table 1. The techniques for exosome isolation
Since liquid biopsy surveillance is performed to detect cancer biomarkers in body fluids, such as blood and urine, the table is missing important advancements in the isolation of EVs:
Find more in Liangsupree, T., Multia, E., & Riekkola, M. L. (2021). Modern isolation and separation techniques for extracellular vesicles. Journal of Chromatography A, 1636, 461773.
e.g.,_
Size and density is lacking Flow field-flow fractionation
Immune specificity is missing completely immunoaffinity chromatography with polymeric monolithic disks, which is a more efficient method for isolation compared to the methods listed in the table.
Review is also lacking Electrophoresis and dielectrophoresis-based isolation methods as well as other charge-based separation methods.
Line 80: Exosome detection approaches based on microfluidics include fluorescence imaging-> quartz crystal microbalance is missing. In addition, there is no mention of Raman spectroscopy.
lines 107-109: A previous study identified two exosome subpopulations via asymmetric-flow 107 field-flow fractionation (AF4), proving that AF4 can be used as an enhanced analytical tool for isolating and addressing the complexities of heterogeneous nanoparticle subpopulations -> not only analytical tool. AF4 can be used in a clinical setting to produce subpopulations of EVs from human plasma with an integrated approach together with immunoaffinity chromatography:
Multia, Evgen, et al. "Automated on-line isolation and fractionation system for nanosized biomacromolecules from human plasma." Analytical chemistry 92.19 (2020): 13058-13065.
The isolation section of the review needs to be improved significantly, while the following chapters don't raise any immediate comments. The authors should also consider replacing exosome with extracellular vesicle.
Round 2
Reviewer 1 Report
The authors have satisfactorily addressed all of my concerns.
Reviewer 2 Report
The authors have done good work and improved significantly the original manuscript based on the comments.